# Banyan: Improved Representation Learning with Explicit Structure

Mattia Opper [1]    N. Siddharth [1]

## Abstract

We present Banyan, a model that efficiently learns semantic representations by leveraging explicit hierarchical structure. While transformers excel at scale, they struggle in low-resource settings. Conversely recent structured models have shown promise as efficient learners, but lack performance. Banyan bridges this gap with two key innovations: an entangled hierarchical tree structure and diagonalised message passing, enabling it to outperform larger transformer models with just 14 non-embedding parameters. It excels in low-resource settings, offering a viable alternative for under-represented languages and highlighting its potential for efficient, interpretable NLP in resource-constrained environments.

## 1. Introduction

Semantic representations are foundational for various NLP applications, such as retrieval-augmented generation (RAG) (Lewis et al., 2020), question answering, and summarisation (Abdalla et al., 2023; Wang et al., 2022). They are also crucial for clustering and organising textual data when labelled training data is unavailable. Typically, such representations are generated by large-scale transformer models (Vaswani et al., 2017); highly effective but needing substantial amounts of data and computational resources to train.

An alternative approach draws inspiration from linguistics and cognitive science, incorporating structured compositions—a principle that posits that understanding the semantics of a whole requires knowing the meanings of its parts and the structural rules that determine how they assemble (Chomsky, 1956; Crain & Nakayama, 1987; Pallier et al., 2011; de Marneffe et al., 2006). This principle is highly efficient because novel utterances can be decomposed into familiar components using systematic rules, minimis-

ing the need to store individual meanings. It allows humans to learn efficiently from relatively little data and enables effective and efficient learners (Lake et al., 2016; Ito et al., 2022; Wiedemer et al., 2023).

To incorporate such inductive biases into models, the traditional information flow within neural networks needs altering. Instead of relying on implicit processing alone, models must learn representations for atomic components and an explicit computation graph that dictates how these components combine. Additionally, models must learn functions to govern information flow through this graph. Such approaches have demonstrated improved language modelling perplexity at cognitively plausible scales (Hu et al., 2021; 2022), better systematic generalisation (Sartran et al., 2022; Murty et al., 2023), and, especially relevant here, enhanced efficiency in acquiring semantics (Opper et al., 2023b).

The SELF-STRAE model (Opper et al., 2023b) learns representations that explicitly model compositional semantics, and achieves promising performance while requiring minimal resources, both in terms of data and model size. It opened the door to exploring more compute-efficient solutions—particularly valuable for low-resource languages where scaling is often infeasible. However, while innovative, it still falls short compared to large-scale pre-trained transformers, even in languages outside standard pre-training corpora. Here, we introduce BANYAN, a model which significantly outperforms SELF-STRAE while achieving greater resource efficiency. Our approach involves modifying the structural optimisation process to induce an *entangled* graph that models global relations between nodes and employs a message passing mechanism using diagonal functions, reducing parameters while enhancing expressiveness.

BANYAN, achieves performance comparable to transformer-based baselines and represents a low-cost, viable alternative to transformers for producing representations in low-resource languages, as measured by semantic textual similarity (STS) tasks. By leveraging cognitively inspired inductive biases, our work enables semantic representation learning that rivals or surpasses large-scale pre-trained LLMs—using only 14 non-embedding parameters. Our model, BANYAN, offers a new direction for efficient and effective semantic understanding in resource-constrained environments. [1]

---

[1]School of Informatics, University of Edinburgh, UK. Correspondence to: Mattia Opper <m.opper@ed.ac.uk>, N. Siddharth <n.siddharth@ed.ac.uk>.

*Proceedings of the 42$^{nd}$ International Conference on Machine Learning*, Vancouver, Canada. PMLR 267, 2025. Copyright 2025 by the author(s).

---

[1]Code available at: github.com/exlab-research/Banyan

## 2. Background and Related Work

Banyan is a graph neural network, specifically a recursive neural network (RvNN) that learns both structure and representations. We unpack these components below.

**Recursive Neural Networks (RvNNs)**: Like regular recurrent neural networks (RNNs), RvNNs process data by repeatedly applying a function to update their state in sequence. However, instead of relying on temporal ordering (like the sequence of words in a sentence), RvNNs use hierarchical structures, often provided as input—most commonly as a binary tree—and can be applied either bottom-up (from leaves to root) or top-down (root to leaves). They were popularised in the deep learning era by Socher et al. (2011; 2013), inspiring many successor models that vary in how they define the recursive function, including Tree-LSTMs (Tai et al., 2015) and IORNN (Le & Zuidema, 2014).

**Learning Structure**: RvNNs often require structural input, which limits their flexibility since this structure may not always be available or easily obtainable. To address this, researchers have developed methods to induce structure within the model during recursive computation; using differentiable chart parsing (Drozdov et al., 2019; 2020; Hu et al., 2021; 2022), beam search (Ray Chowdhury & Caragea, 2023), continuous relaxation (Chowdhury & Caragea, 2021; Soulos et al., 2024), and reinforcement learning (Havrylov et al., 2019). However, these methods can struggle with memory issues and require careful tuning of hyperparameters. Here, we adopt a method from Opper et al. (2023b) that uses representation similarity to determine how nodes should be merged during computation, which is both computationally efficient and surprisingly effective.

**Semantic Representations of Text**: Systems like Word2Vec (Mikolov et al., 2013) and GloVe (Pennington et al., 2014) use the distributional hypothesis (Harris, 1954) to model word semantics, which posits that words are defined by the context in which they appear. To learn representations, these models use a fixed context window and predict a missing word in a sequence. While initially effective, this approach is limited because representations for higher level objects (i.e. phrases, sentences etc.) are computed by simply averaging word embeddings. However, some notable follow on works attempted to improve upon this. Arora et al. (2017); Ethayarajh (2018) introduce a more sophisticated form of taking an average over word embeddings by using SVD, while Rücklé et al. (2018) use the power mean. These approaches yielded improvements, but relied on representations pre-trained at scale which limited their applicability to specialised domains or low resource languages. Wieting et al. (2021) attempt to refine the sentence representation through the use of paraphrase corpora, looking to increase alignment between language pairs, again improving performance, but requiring large scale parallel corpora. Finally,

Pagliardini et al. (2017) realised that if the average of the embeddings was going to be used to create the sentence representation, it makes sense to optimise it directly. Consequently, they modified the pre-training objective in order to have the average predict a missing word from a sentence. At scale this proved tremendously effective. However, despite offering substantial improvements, all these methods require scale and more importantly do not tackle the central limitation of word embeddings - the inability to handle changes in meaning dependent on context.

On the other hand, transformers, through self-attention, are able to represent contextualised meanings. However, early encoder-only transformer models produced poor representations (Reimers & Gurevych, 2019), especially compared to the more sophisticated approaches based on word embeddings. This was largely due to the anisotropy issue (Godey et al., 2024), which required the development of techniques using contrastive fine-tuning (Gao et al., 2021) to finally remedy. These approaches eventually surpassed word embeddings, and have become the method of choice for producing semantic representations. However, they still rely heavily on scale for success, as contrastive refinement is a final fine-tuning step applied to pre-trained models rather than directly incorporated within pre-training.

**Semantic Representation Learning through Structure**: Transformer embeddings have become more successful than static word embeddings due to their ability to handle varying contextual influences. Unlike attention mechanisms in transformers, which route information based on token relationships, some approaches use explicit graphs or structures, such as dependency (Levy & Goldberg, 2014; Vashishth et al., 2019) or constituency parses (Pham et al., 2015), to determine the focus of context windows. These models have the potential to bridge the gap between the efficiency of word embeddings and the contextualisation offered by transformers. This is because the discrete structure provides an input specific routing order which dictates interactions between atoms and consequently determines their influence on higher level representations - allowing for more flexibility than simple averaging. Most related to our work, Opper et al. (2023b) introduce two models. StrAE, which use constituency parsers to learn sentence-level embeddings alongside word embeddings, and SELF-STRAE, which learns its own structure using representations. This latter model, SELF-STRAE, serves as the foundation for BANYAN and is described next.

## 3. Preliminary: Self-StrAE

SELF-STRAE involves three main components that act over a sequence of tokens $\mathbf{w} = \langle w_n \rangle_{n=1}^{N}$: (a) an algorithm for merging tokens based on their similarity, (b) functions for composition and decomposition of embeddings, and (c) an

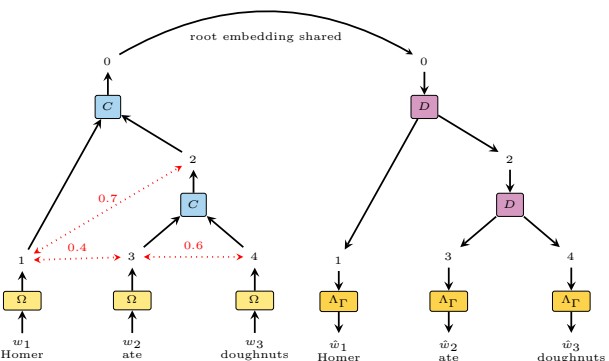

Figure 1: Self-StrAE operation. Red lines indicate cosine similarity. Shared colours imply shared parameters.

objective that leverages both the induced structure and embeddings. While full details are available in Opper et al. (2023b), we provide a brief overview to establish context for our model development (§ 4).

At a high level, SELF-STRAE learns representations that define their own structure while being shaped by it. Starting with an initial embedding matrix $\Omega_\Psi$, tokens are merged into single embeddings using a composition function $C_\Phi$ based on best cosine similarity (e.g., see Figure 1). This process reduces the sequence to a single root embedding while capturing semantic relationships. The resulting merge history forms a binary tree structure, over which the model then operates in reverse by decomposing embeddings at each node using a decomposition function $D_\Theta$, to reconstruct the leaf embeddings. Optionally, it can further predict tokens ($\widehat{w}_n$) using a dembedding function $\Lambda_\Gamma$. Figure 1 illustrates the autoencoding process. During training, tokens that are frequently merged together develop correlated representations, leading the model to learn meaningful compositional semantics. This results in embeddings that reflect both their own structure and the semantic patterns they encode.

More formally, one denotes tokens as the vertices $w_i \in \Delta^V$ in a $V$-simplex for vocabulary size $V$, and note that the models generates two sets of embeddings—one going up ($\bar{e}$: leaves $\rightarrow$ root) and one coming down ($\underline{e}$: root $\rightarrow$ leaves). The embeddings are viewed as $e \in \mathbb{R}^{U \times K}$ allowing the composition and decomposition functions to act independently over $K$ channels, and be defined as

$$C_\Phi(\bar{e}_i, \bar{e}_{i+1}) = \text{HCAT}(\bar{e}_i, \bar{e}_{i+1})\,\Phi + \phi, \quad \Phi \in \mathbb{R}^{2U \times U} \quad (1)$$

$$D_\Theta(\underline{e}_i) = \text{HSPLIT}(\underline{e}_i\,\Theta + \theta), \qquad \Theta \in \mathbb{R}^{U \times 2U} \quad (2)$$

To learn this model from data, Opper et al. (2023b) derive two objectives. The first is straightforward cross-entropy over reconstructed tokens, for sentence $\mathbf{w}$ and prediction $\widehat{\mathbf{w}}$ as $\mathcal{L}_{\text{CE}}(\mathbf{w}, \widehat{\mathbf{w}}) = -\frac{1}{N}\sum_{n=1}^{N} w_n \cdot \log \widehat{w}_n$. This however, places little constraint on the intermediate nodes in the hierarchical model. To address this, an alternate structural

contrastive objective is formulated over a batch of sentences. As up and down trees are structurally identical (modulo edge reversal), it draws together an embedding and its dual on the other tree, while pushing away all other embeddings across the batch, using cosine similarity. Denoting pairwise similarity matrix $A \in \mathbb{R}^{M \times M}$ between up and down embeddings over $M$ nodes in the batch, the objective is: $\mathcal{L}_{\text{CO}}(\bar{e}, \underline{e}) = \frac{-1}{2M}\sum_{i=1}^{M} \log\left(\sigma_\tau(A_{i\cdot})\,\sigma_\tau(A_{\cdot i})\right)$ with $\sigma_\tau(\cdot)$ the tempered softmax over the unspecified $(\cdot)$ dimension.

# 4. Banyan

Given their construction, the upward embeddings are always *locally-contextual*: only encapsulating the context of the span they cover. For example, in Figure 1, the upward embedding $\bar{e}$ for the span "ate doughnuts" is always the same regardless of context, no matter who did the eating. In contrast, downward embeddings are always *globally-contextual*: necessarily encapsulating surrounding context, being decomposed from embeddings of larger spans. In our example, this implies multiple downward embeddings $\underline{e}^y$, one for each $y \in \{\text{"Lisa"}, \text{"Homer"}, \dots\}$. Learning effective embeddings requires amortisation over these differences to ensure meaning resolves over all these contexts.

## 4.1. From trees to entangled trees

---

**Algorithm 1** BANYAN: Entangled Compose

---

**Input:** Global frontier $\langle(s_n, e_n)\rangle_{n=1}^N$, compose ($\circ$), concat ($\diamond$), similarity $\text{CSIM}(e, e')$

1:  $\mathcal{A} \leftarrow \langle(s_n, e_n)\rangle_{n=1}^N$        ▷ initialise frontier
2:  $(\mathcal{V}, \mathcal{E}) \leftarrow (\varnothing, \varnothing)$        ▷ initialise graph
3:  **while** $\exists i : s_i \diamond s_{i+1} \notin_s \mathcal{V}$ **do**
4:     $i^\star \leftarrow \arg\max_i \text{CSIM}(e_i, e_{i+1})$    ▷ locate closest pair
5:     $e_p = \circ(e_{i^\star}, e_{i^\star+1})$        ▷ compose
6:     $\mathcal{V} \leftarrow \mathcal{V} \cup \{(s_{i^\star} \diamond s_{i^\star+1}, e_p)\}$
7:     $\mathcal{E} \leftarrow \mathcal{E} \cup \{p \sim i^\star, p \sim (i^\star + 1)\}$
8:     $\mathcal{J} \leftarrow \{j : (s_j, s_{j+1}) = (s_{i^\star}, s_{i^\star+1})\}$
              ▷ locate all occurrences of this pair
9:     $\mathcal{A} \leftarrow \mathcal{A} \setminus \{\forall_{j \in \mathcal{J}}\ \mathcal{A}_j, \mathcal{A}_{j+1}\}$
              ▷ delete occurrences from those locations
10:    $\mathcal{A} \leftarrow \mathcal{A} \cup_\mathcal{J} \{(s_{i^\star} \diamond s_{i^\star+1}, e_p)\}$
              ▷ insert composition into those locations

**return:** Graph $(\mathcal{V}, \mathcal{E})$

---

We wish to have composition embeddings amortise over all possible contexts, and simultaneously, all decompositions embeddings to resolve to the same thing. The representation of an entity "Lisa" should encapsulate everything she could possibly eat. Simultaneously, the average of everything she could eat we should get back to "Lisa". Self-StrAE does not explicitly model this behaviour in its structure. Decomposition embeddings of the same entity only interact

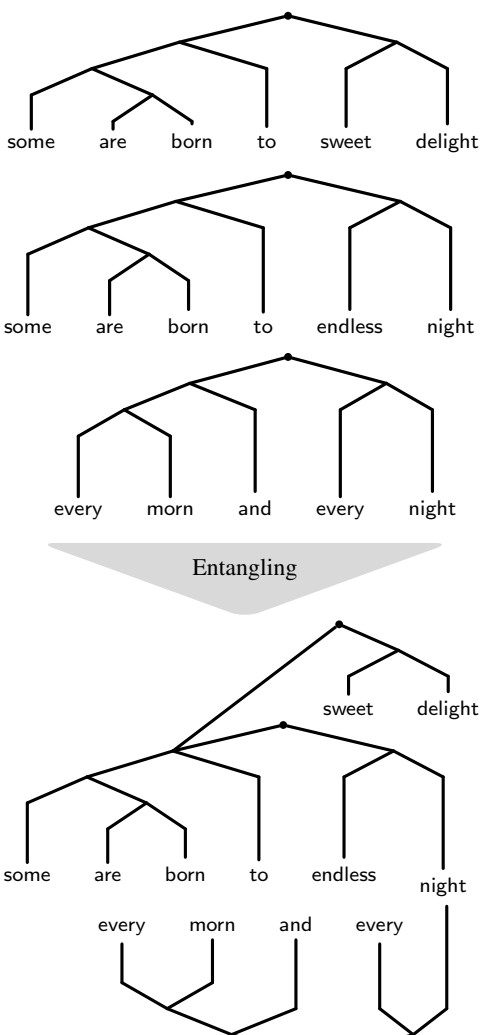

Figure 2: Entangled trees: Example of disjoint trees being transformed into an entangled tree. Internal functions $(C_\Phi, D_\Theta, \dots)$ are elided to avoid clutter.

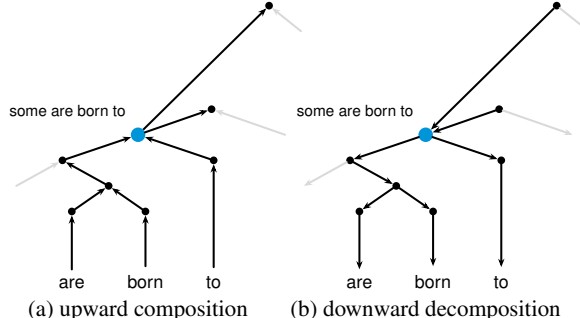

(a) upward composition    (b) downward decomposition

Figure 3: Upward and downward traversals for a section of the entangled tree from Figure 2.

when we calculate the loss. On top of this, because the loss is taken over the batch, they are actually treated as false negatives to each other. Even though they are terms that ought not be pushed away, the objective ask them to be.

Our innovation is to address both these issues by formulating the process in terms of entangled trees—where entangling describes the transformation of disjoint tree structures into a conjoined graph structure. An example is shown in Figure 2. Here, all instances of "night" and "some are born to" are captured by a single node representing that constituent. We call our model BANYAN on account of this entangling, because, like the tree, it can have many roots—consisting of nodes frequently reused across contexts.

**Entangling:** Constructing an entangled tree given a set of disjoint trees is a relatively straightforward process and is formally specified in Algorithm 1. In contrast to the

agglomerative clustering employed in SELF-STRAE, here we employ a global frontier spanning all leaf nodes across the given data. The key differences to the prior methods are mainly to do with constructing a graph jointly with progressing the frontier and ensuring that new nodes are never duplicated, for which we employ a node identity $s_n$ in addition to the node embedding $e_n$.

**Incorporating context:** Following the entangling of trees described, the model proceeds in a similar vein to SELF-STRAE, by composing upwards from leaves to roots (multiple roots corresponding to multiple trees), and then decomposing downwards back to the leaves. With entangled trees, while traversing upwards each node is always composed from the same two children, but on the way back down, things are different as each separate context for a given node provides a different downward embedding. This is shown in Figure 3 focussing on a subgraph of the entangled tree from Figure 2(right). Note that the node in question (in blue) corresponds to the span "some are born to", and has downward embeddings that incorporate context both from "endless night" and "sweet delight". This is exactly as desired, as BANYAN allows explicit aggregation to derive the downward embedding that resolves over the contexts. For any upward embedding $\bar{e}$ whose span occurs in different contexts $y \in \mathcal{Y}$, the corresponding downward embedding is derived by simply averaging over the different contextual down embeddings; i.e., $\underline{e} = 1/|\mathcal{Y}| \sum_y \underline{e}^y$.

**Effectiveness and efficiency:** Beyond the ability to explicitly incorporate context across data, entangled trees also help the contrastive objective by avoiding false negatives since they do not admit duplicate nodes by construction. Furthermore, the lack of duplicate nodes also drastically impacts the memory footprint of the model as one deals with the *set* of all nodes rather than counting each instance as its own node. These effects becomes more pronounced when entangling a larger set of instances as the likelihood of false negative and duplicates goes up together.

**Practical estimation:** Given the advantages of entangled trees, one would ideally want to construct it over *all* the

available data—not practically feasible with the exponential growth in dataset sizes. To address this, we construct our model to estimate the given objective by taking steps over *batches* of data that are of a more manageable size, noting that this estimator is unbiased. To see this is the case, note that entangled trees only affects the downward embeddings directly, and that batching simply means that the resolved embedding is an average over *samples* instead of over all the data (*population*)—the sample mean is always an unbiased estimator of the population mean.

## 4.2. Simplified Message Passing

Complementary to the development of entangled trees to incorporate context, we also explore avenues to improve the message passing with the composition ($C$) and decomposition ($D$) functions. The original formulations (1, 2) concatenate or split using simple single-layer linear neural networks. These were found to lead to better representations than e.g., Tree-LSTM cells, because they forced the model to conform to the compression order of the structure.

But if all that was required for success is to respect the compression order, then one could possibly do better with a simpler solution that exploits diagonalised functions (Ba et al., 2016)—a crucial component in the resurgence of recurrent neural networks (Peng et al., 2023; Orvieto et al., 2023; De et al., 2024) introducing decayed memory across time. Thus, rather than using linear layers, we now define:

$$C(\bar{e}_i, \bar{e}_{i+1}) = (\bar{e}_i \cdot \sigma(\Phi_l) + \bar{e}_{i+1} \cdot \sigma(\Phi_r)) + \phi \quad (3)$$

$$D(\underline{e}_i) = \left(\underline{e}_i \cdot \sigma(\Theta_l) + \theta_l, \ \underline{e}_i \cdot \sigma(\Theta_r) + \theta_r\right) \quad (4)$$

$$\Phi_l, \Phi_r, \phi, \Theta_l, \Theta_r, \theta_l, \theta_r \in \mathbb{R}^U$$

with sigmoid non-linearity ($\sigma$) applied to parameters both for numerical stability and to enforce a decayed memory over structure depth. Repeated application of the diagonal composition function will decay the influence of nodes further down in the tree, thereby respecting its compression order. During composition representations can increase in magnitude as they are the sum of the two children. During decomposition representations will, by necessity, reduce back down in magnitude towards the leaves. Further mimicking the information flow specified by the entangled trees. Finally, they restrict encoder (comp) and decoder (decomp) embeddings to remain in the same space. Which makes amortisation required for successful reconstruction, letting us switch objective to **cross entropy** over the vocabulary. We provide analysis to support this claim later in § 7.

These relatively simple changes have a pretty drastic effect, both in terms of performance (see experiments), as well as efficiency, with parameters now reduced by a factor of $U$ compared to the functions from (1, 2).

## 5. Experiments: English Evaluation

**Goal:** We wish to test whether BANYAN can efficiently learn semantics. We start by evaluating on English, which is well resourced and has a wide array of test sets available with which we can measure the efficacy of our embeddings. This is crucial to establish, because when we turn to low resource languages later on, the amount of reliable evaluation sets will become limited. We want to make use of broad spectrum of tests available for English to reliably demonstrate embedding quality before moving forward.

### 5.1. Experimental Setup and Evaluation:

We want to evaluate how well BANYAN learns effective semantic representations. Ideally we want to probe this at different levels of hierarchy, because it allows us to test whether structured models can do what they are supposed to i.e., seamlessly transfer semantic knowledge across different levels of hierarchy via composition. Our evaluation is unsupervised, both to directly probe the effect of the inductive bias, and for greater parity with what may be expected in a low-resource domain. It consists of three parts:

**Correlation with human judgements:** We compare the cosine similarity of embedding pairs produced by the model with human judgements of their semantic correspondence. On the word level, we use Simlex-999 (Hill et al., 2015) and WordSim-S/R (Agirre et al., 2009). All tasks measure semantics, but do so on differing axes. Simlex measures similarity at the exclusion of relatedness. Wordsim S measures similarity without penalising relatedness. And Wordsim R measures relatedness. On the sentence level, we use STS-12 through 16 (Agirre et al., 2012; 2013; 2014; 2015; 2016), the STS-B (Cer et al., 2017), SICK-R (Marelli et al., 2014) and SemRel (Ousidhoum et al., 2024) - which combined cover a wide array of semantic correspondence.

**Retrieval:** This is a cornerstone of Retrieval-Augmented Generation (RAG)-based systems and perhaps the most important use case for embedding models. We use two retrieval datasets from the BEIR suite (Thakur et al., 2021). Quora: evaluates success of matching questions to answers and capturing the response relation. Arguana: evaluates matching arguments to counter arguments, testing if our semantic space captures the notion of dialectical opposition.

**Classification:** We also include two test sets from the GLUE benchmark (Wang et al., 2019). Sentiment classification (SST-2) tests whether the representation space captures semantic polarity. Paraphrase detection (MRPC): tests whether our representation space capture semantic equivalence. While our other evaluation is applied to the embeddings zero-shot, for the classification tasks we train a GeLU MLP with a 512D hidden size, though we leave models frozen as a direct test of representation quality.

Table 1: Sentence level results for models pretrained on English. Higher is better. Results represent mean and standard deviation across four random initialisations. Spearman's $\rho$ is * 100 following convention.

| Model | STS-12 | STS-13 | STS-14 | STS-15 | STS-16 | STS-B | SICK-R | SemRel | Score |
|---|---|---|---|---|---|---|---|---|---|
| SELF-STRAE | $31.98 \pm 0.58$ | $53.88 \pm 0.68$ | $37.73 \pm 0.70$ | $55.23 \pm 0.58$ | $55.55 \pm 0.47$ | $39.53 \pm 1.61$ | $51.78 \pm 0.29$ | $50.05 \pm 0.92$ | $46.59 \pm 0.43$ |
| GLOVE | $31.61 \pm 0.31$ | $21.69 \pm 0.12$ | $27.37 \pm 0.10$ | $40.42 \pm 0.09$ | $29.27 \pm 0.12$ | $28.25 \pm 0.08$ | $50.20 \pm 0.25$ | $41.20 \pm 0.43$ | $33.75 \pm 0.04$ |
| + stopword rm | $39.00 \pm 0.57$ | $41.61 \pm 0.19$ | $39.31 \pm 0.18$ | $51.06 \pm 0.35$ | $45.14 \pm 0.14$ | $48.40 \pm 0.07$ | $52.80 \pm 0.04$ | $42.37 \pm 0.13$ | $44.96 \pm 0.10$ |
| Sent2Vec | $38.14 \pm 0.29$ | $51.37 \pm 0.48$ | $48.64 \pm 0.09$ | $67.28 \pm 0.02$ | $56.26 \pm 0.06$ | $53.39 \pm 0.11$ | $\mathbf{59.67 \pm 0.02}$ | $51.47 \pm 0.03$ | $53.28 \pm 0.11$ |
| ROBERTA | $42.77 \pm 1.27$ | $51.70 \pm 1.30$ | $45.67 \pm 1.42$ | $63.97 \pm 0.81$ | $59.60 \pm 0.61$ | $39.97 \pm 0.95$ | $52.93 \pm 0.23$ | $52.73 \pm 0.58$ | $51.08 \pm 0.61$ |
| + SimCSE | $\mathbf{50.63 \pm 1.45}$ | $62.23 \pm 2.51$ | $54.17 \pm 2.10$ | $68.77 \pm 3.00$ | $\mathbf{66.67 \pm 1.40}$ | $53.53 \pm 1.18$ | $56.87 \pm 1.16$ | $59.27 \pm 0.93$ | $59.02 \pm 1.45$ |
| BANYAN | $\mathbf{51.38 \pm 0.15}$ | $\mathbf{69.60 \pm 0.37}$ | $\mathbf{63.20 \pm 0.28}$ | $\mathbf{73.08 \pm 0.26}$ | $67.18 \pm 0.56$ | $\mathbf{61.90 \pm 0.63}$ | $55.23 \pm 0.13$ | $\mathbf{61.88 \pm 0.22}$ | $\mathbf{62.97 \pm 0.03}$ |

Table 2: Word level results analogous to Table 1.

| Model | Simlex | Wordsim-S | Wordsim-R | Score |
|---|---|---|---|---|
| SELF-STRAE | $13.80 \pm 0.41$ | $54.38 \pm 0.78$ | $52.85 \pm 1.27$ | $40.34 \pm 0.66$ |
| GLOVE | $27.47 \pm 0.25$ | $62.53 \pm 0.42$ | $51.00 \pm 0.56$ | $47.00 \pm 0.38$ |
| Sent2Vec | $\mathbf{28.88 \pm 0.42}$ | $\mathbf{68.32 \pm 1.28}$ | $54.49 \pm 1.51$ | $\mathbf{50.56 \pm 0.79}$ |
| ROBERTA | $\mathbf{29.23 \pm 0.64}$ | $61.97 \pm 2.38$ | $46.00 \pm 2.13$ | $45.73 \pm 1.71$ |
| BANYAN | $14.65 \pm 2.90$ | $63.23 \pm 2.21$ | $\mathbf{67.73 \pm 0.3}$ | $\mathbf{48.53 \pm 1.33}$ |

**Baselines:** We compare against the SELF-STRAE, GLOVE (Pennington et al., 2014), Sent2Vec (Pagliardini et al., 2017) and a ROBERTA (Liu et al., 2019) in the medium configuration from (Turc et al., 2019; Opper et al., 2023a). SELF-STRAE, the closest point of comparison to BANYAN, indicates where the current performance level of structured representation learning lies. GLOVE lets us compare to traditional static embeddings, and tests whether our model is learning anything more than just simple bag of word features. To obtain sentence embeddings, we report results using both the simple average of the word embeddings and the average with stopwords removed following (Reimers & Gurevych, 2019). The latter is generally stronger. An even more powerful variant is Sent2Vec, which directly optimises the averaged representation by using it to represent the context. This comparison measures the utility BANYAN's flexible and parametrised composition process, compared to an optimised average. Finally, for ROBERTA, we report results using both the standard model, and again after enhancing ROBERTA through an extra round of contrastive SIMCSE training (Gao et al., 2021), as a further non-lexical baseline. Our pooling strategy is mean. To produce static embeddings from ROBERTA to use in lexical evaluation, we follow Bommasani et al. (2020) and average the contextualised representations of all occurrences of the word in the training set. The ROBERTA is intended as a stronger baseline. It has significantly more parameters than BANYAN and can model meaning in context unlike GLOVE and Sent2Vec.

**Hyperparameters and Pre-Training Details:** For all models we set the embedding size to 256. For SELF-STRAE we use the configuration of (Opper et al., 2023b) and set embeddings as square matrices (i.e., $K$=16 and $U$=16). For BANYAN we set these values to $K$=128 and $U$=2, because the more independent channels we allowed the better the

model seemed to perform. We refer the reader to Appendix A for ablations. We also note that because we can perform this reduction in channel size, the number of non-embedding parameters for BANYAN drops to just 14, as these are directly proportional to $U$. The configuration for RoBERTa medium is 8 layers, 8 attention heads, 2048 dimensional feed forward layers, and relative positional embeddings. For SELF-STRAE, BANYAN and Sent2Vec we pre-train on a uniform subsample of English Wikipedia consisting of circa 10 million tokens. This represents the lower end of how many tokens might be available in a low resource setting, and allows us to test whether these methods are efficient. Meanwhile for GLOVE and ROBERTA we pre-train on Wiki-103 (Merity et al., 2016), to ensure that they are not penalised by insufficient scale. Wiki-103 comprises circa 100 million tokens and therefore represents the very upper end of what might be available in a low resource setting. Further training details are in Appendix B.

### 5.2. Results:

Results are shown in Tables 1 to 3. On both the word level and sentence level BANYAN does much better than SELF-STRAE. We ablate the reasons for this in more detail later in the manuscript. Both models suffer on SimLex because they need to model both similarity and relatedness as the latter dictates merge (related concepts often compose together). However, the important thing to note is that the structured models effectively transfer the same performance from the word level to the sentence level. They can take advantage of composition, and transfer the meaning of the parts to understanding the meaning of the whole. The GLOVE baseline is good on the word level, but does not generalise to the sentence level as well as the transformer, even when we give it stopword removal. Similarly, Sent2Vec is extremely strong on the word level, and while more effective than GLOVE, neither approach *seamlessly* transfers semantic knowledge to different levels of complexity. BANYAN can, and is able to achieve comparable or better performance than the SimCSE ROBERTA despite being much smaller and exposed to 10x less pre-training data. This means we have a structured model that remains efficient and cheap, and also effective at representation learning.

Table 3: Sentence level results on retrieval and classification tasks for models pretrained on English.

| | Quora | | | | Arguana | | | | SST-2 | MRPC |
|---|---|---|---|---|---|---|---|---|---|---|
| Model | NDCG@1 | NDCG@10 | R@1 | R@10 | NDCG@1 | NDCG@10 | R@1 | R@10 | Acc | F1 |
| Self-StrAE | 32.88 ± 0.28 | 40.02 ± 4.94 | 29.59 ± 0.23 | 44.77 ± 0.28 | 09.96 ± 0.11 | 15.48 ± 0.13 | 09.96 ± 0.11 | 21.48 ± 0.22 | 74.67 ± 0.52 | 80.34 ± 0.42 |
| GloVe | 29.99 ± 0.14 | 35.71 ± 0.15 | 26.08 ± 0.15 | 43.17 ± 0.25 | 06.19 ± 0.19 | 12.77 ± 0.24 | 06.18 ± 0.19 | 24.68 ± 7.40 | 75.83 ± 0.62 | 81 ± 0 |
| + stopword rm | 44.41 ± 0.13 | 52.54 ± 0.17 | 38.78 ± 0.15 | 62.15 ± 0.25 | 09.89 ± 0.19 | 20.27 ± 0.09 | 09.89 ± 0.19 | 33.00 ± 0.26 | 76.50 ± 1.08 | 81 ± 0 |
| Sent2Vec | 36.12 ± 0.21 | 43.26 ± 0.15 | 31.33 ± 0.21 | 52.38 ± 0.05 | 09.60 ± 0.31 | 23.24 ± 0.15 | 09.60 ± 0.31 | 39.73 ± 0.89 | 76.53 ± 0.98 | 81 ± 0 |
| RoBERTa | 43.26 ± 0.76 | 49.97 ± 0.72 | 37.67 ± 0.68 | 58.78 ± 0.78 | 08.18 ± 0.43 | 17.60 ± 0.36 | 08.18 ± 0.43 | 28.85 ± 0.94 | 75.68 ± 0.96 | 81 ± 0 |
| + SimCSE | 51.79 ± 2.12 | 59.30 ± 2.10 | 45.09 ± 1.60 | 68.74 ± 2.01 | 10.06 ± 1.27 | 21.84 ± 2.23 | 10.06 ± 1.27 | 37.36 ± 2.16 | 75.97 ± 1.08 | 80.83 ± 0.24 |
| Banyan | **57.74 ± 0.10** | **65.71 ± 0.14** | **50.14 ± 0.10** | **75.71 ± 0.14** | **12.42 ± 0.40** | **28.28 ± 0.17** | **12.42 ± 0.40** | 48.19 ± 0.18 | 75.96 ± 0.57 | 79.48 ± 0.42 |

Table 4: Multilingual Results. BANYAN performance over four random seeds. Baselines marked † finetuned on supervised semantic similarity datasets. FT–unsupervised finetuning using masked language modelling on same corpora as BANYAN.

| Model | Indonesian | Arabic | Telugu | Marathi | Hausa | Afrikaans | Spanish | Amharic | Hindi | Score |
|---|---|---|---|---|---|---|---|---|---|---|
| XLM-R | 46.7 | 31.6 | 46.3 | 55.7 | 4.1 | 56.2 | 68.9 | 57.3 | 52.7 | 46.61 |
| Llama-3.1 (8B) | **53.4** | 31.1 | 65.6 | 63.4 | 6.1 | 65.4 | 66.7 | 64.1 | **61.7** | 53.06 |
| Mistral Nemo | 50.7 | 20.1 | 57 | 52.3 | 1.8 | 58.3 | 66.2 | 53.2 | 55.8 | 46.16 |
| MiniLM-L12† | 39 | 16.1 | 34.8 | 39.5 | 32.7 | 74.1 | 58.8 | 9.6 | 43.8 | 38.71 |
| Paraphrase XLM-R† | 46.1 | **61** | 58.1 | **79.6** | 22.5 | 76.8 | 71.7 | 64.6 | 52 | 59.16 |
| XLM-R (FT) | 47.9 | 33.6 | 68.8 | 75.1 | 14.6 | 72.6 | **72.8** | 59.6 | 57.6 | 55.84 |
| BANYAN | 41.90 ± 0.56 | 42.28 ± 1.57 | **71.58 ± 1.24** | 66.38 ± 0.84 | **49.68 ± 0.75** | **79.35 ± 0.65** | 60.88 ± 0.86 | **66.40 ± 0.90** | **61.63 ± 0.43** | **60.01 ± 0.35** |

# 6. Experiments: Multilingual Evaluation

**Goal:** We've established that BANYAN is an efficient learner. This implies potential use for languages that are not well covered by current NLP approaches. Here we test that.

## 6.1. Experimental Setup and Evaluation:

**Tasks:** Learning semantic representations for low-resource languages remains an ongoing challenge. A core problem is not just the lack of training data, but also the lack of evaluation datasets. Recent work by Ousidhoum et al. (2024) has sought to address this issue, providing semantic relatedness test sets for several low resource Asian and African languages, evaluated by comparing embedding similarity to human judgements. This means we can BANYAN's ability on Indonesian, Arabic, Telugu, Marathi, Hausa, Afrikaans, Spanish, Amharic and Hindi—covering a broad spectrum of resource extent. For example, Spanish is generally well represented, while Hausa is considerably less so.

**Baselines:** We select XLM-R (Conneau et al., 2019): a transformer encoder trained on 2TB of multilingual data. Llama 3.1 8B (Dubey et al., 2024): a decoder only LLM trained on 15 trillion tokens. Mistral Nemo 12B: a decoder only LLM designed with multi-lingual capacities in mind. In addition we also compare against two specialised embedding models from the sentence transformers range (Reimers & Gurevych, 2019): Mini-LM-L12-V2 and Paraphrase-XLM-R-Multilingual-V1. These are pre-trained encoders

that have been finetuned on supervised datasets designed to produce high quality semantic representations. The baselines we select here are emblematic of the kind of models one might reach for in order to embed a corpus. For all baselines we use mean pooling following (Reimers & Gurevych, 2019). Finally, for parity we include an XLM-R baseline which is finetuned on the same corpora as BANYAN.

**Banyan Pre-training and Hyperparameters:** For Afrikaans, Spanish and Amharic we obtained corpora from Leipzig Corpora Collection (Goldhahn et al., 2012). For Amharic we utilised a MIT licenced pre-training set of 1 million sequences available at this link. Hausa data was sourced from Opus (Nygaard & Tiedemann, 2003). Each dataset consists of roughly 10 million tokens. We utilise a pretrained BPE tokeniser for each language from the BPEMB Python package (Heinzerling & Strube, 2018). BANYAN's hyperparameters remain the same as before. For XLM-R we finetune for 100k steps with early stopping, using a linearly scheduled learning rate of 5e-5 with 10% of steps as warmup. XLM-R runs at batch size 128 across 4xA40 cards.

## 6.2. Results

See Table 4. In Spanish, a well-resourced language with high coverage, the transformer baselines almost all outperform BANYAN. However, as languages become lower resourced the picture changes, and BANYAN outperforms or is comparable to the baselines. This even includes the multilingual XLM-R that has undergone supervised training. While

Table 5: Number of non-embedding parameters.

| Model | BANYAN | SELF-STRAE | ROBERTA (M) | All-MiniLM-L12-V2 | XLM-R | Llama 3.1 | Mistral Nemo |
|---|---|---|---|---|---|---|---|
| Params | 14 | 1072 | ≈10M | ≈21M | ≈85M | ≈8B | ≈12B |

Table 6: Ablations of modelling changes made for Banyan. Higher is better. Results averaged across four random initialisations. Bolded results indicate no standard deviation overlap. Spearman's $\rho$ is * 100 following convention.

| Model | STS-12 | STS-13 | STS-14 | STS-15 | STS-16 | STS-B | SICK-R | SemRel | Score |
|---|---|---|---|---|---|---|---|---|---|
| Standard Trees | $31.98 \pm 0.58$ | $53.88 \pm 0.68$ | $37.73 \pm 0.70$ | $55.23 \pm 0.58$ | $55.55 \pm 0.47$ | $39.53 \pm 1.61$ | $51.78 \pm 0.29$ | $50.05 \pm 0.92$ | $46.59 \pm 0.43$ |
| + diag functions | $35.13 \pm 0.33$ | $56.05 \pm 0.24$ | $40.58 \pm 0.05$ | $58.83 \pm 0.10$ | $56.78 \pm 0.21$ | $44.10 \pm 0.14$ | $53.35 \pm 0.17$ | $52.65 \pm 0.17$ | $49.68 \pm 0.06$ |
| ++ CE loss | $47.10 \pm 1.04$ | $61.85 \pm 1.44$ | $58.60 \pm 1.34$ | $70.45 \pm 0.57$ | $62.45 \pm 0.70$ | $59.50 \pm 0.53$ | $\mathbf{59.00 \pm 0.26}$ | $60.33 \pm 0.26$ | $59.91 \pm 0.54$ |
| Entangled Trees | $38.98 \pm 0.39$ | $61.75 \pm 0.14$ | $43.65 \pm 0.46$ | $58.21 \pm 0.41$ | $55.29 \pm 0.23$ | $46.15 \pm 0.71$ | $53.93 \pm 0.16$ | $52.53 \pm 0.09$ | $51.31 \pm 0.13$ |
| + diag functions | $44.15 \pm 0.002$ | $62.80 \pm 0.002$ | $48.30 \pm 0.001$ | $64.60 \pm 0.002$ | $60.30 \pm 0.001$ | $49.80 \pm 0.002$ | $55.14 \pm 0.001$ | $57.70 \pm 0.001$ | $55.23 \pm 0.01$ |
| ++ CE loss | $\mathbf{51.38 \pm 0.15}$ | $\mathbf{69.60 \pm 0.37}$ | $\mathbf{63.20 \pm 0.28}$ | $\mathbf{73.08 \pm 0.26}$ | $\mathbf{67.18 \pm 0.56}$ | $\mathbf{61.90 \pm 0.63}$ | $55.23 \pm 0.13$ | $\mathbf{61.88 \pm 0.22}$ | $\mathbf{62.97 \pm 0.03}$ |

finetuning XLM-R improves performance the amount of benefit it provides is not uniform and is insufficient to prove viable in the very low resource cases. BANYAN is able to learn competitive representations consistently across languages, unsupervised and with very little data, providing a viable alternative for cheaply and efficient embeddings for low resource languages.

## 7. Improvements and Ablations

### 7.1. Efficiency

Other than its embedding matrix, BANYAN only has composition and decomposition functions. Diagonalising these makes them easier to compute and more lightweight than standard weight matrices, ($2U$ rather than $2U \times U$), achieving a further order of magnitude reduction in parameters compared with the already minimal SELF-STRAE. Table Table 5 shows the difference, including a comparison to the various baselines used throughout the paper. Despite its size BANYAN remains competitive.

Secondly, by exploiting entangled tree structure the number of nodes grows at a significantly reduced rate with batch size compared with standard sentential trees (see Figure 4). This is because the number of reused constituent nodes also grows as batch size increases, and entangled trees capture the set of all constituents, which consequently does not grow as drastically. In practical terms, because entangled trees requires fewer nodes, and each node requires two distinct embeddings ($\bar{e}$ and $\underline{e}$) to be held for it, reducing the number of nodes required leads to radical improvements in memory efficiency. Put together, these changes mean that we can train BANYAN very quickly as we can use large batches and its small number of parameters ensure quick convergence. On a single Nvidia A40 GPU with a batch size of 1024, Banyan trains from scratch in under 50 minutes, meaning that the total cost of pretraining a BANYAN model sits at around 30 cents[2]. Free-tier Google Colab users can achieve

[2]Cloud computing costs from: https://www.runpod.io/pricing

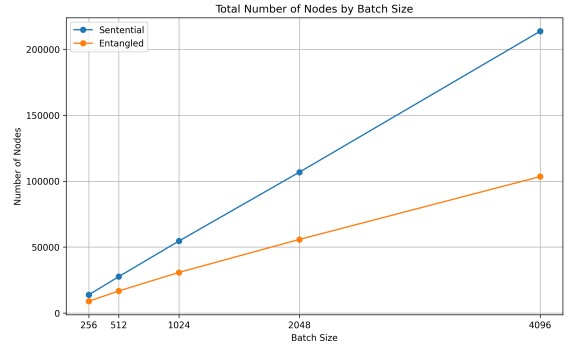

Figure 4: Total number of nodes in entangled vs. sentential trees as batch size grows.

similar results in about two hours with a smaller batch size. Inference can also be performed on CPU on typical laptops, because the model is so small. Combined with its data efficiency, we believe this provides a promising alternative for low resource languages and communities.

### 7.2. Ablations

Why is BANYAN so much more effective than its SELF-STRAE predecessor? To probe the impact of each change, we perform ablations using the English STS tasks (Table 6).

Beyond improving efficiency, changing to entangled trees yields some benefits in terms of performance. The effect is significantly more pronounced when using the contrastive objective, as it removes the issue of false negatives as discussed in Section 4. However, it also yields some slight benefit with the CE leaf reconstruction objective. Entangling explicitly allows the model to take advantage of shared constituency structure between complex sequences, as it combines the information from all incoming parent messages. The slight edge this provides indicates that explicitly allowing the model to take advantage of such systematicity may be useful. However, in terms of performance, we find that the choice of functions and objective plays a much bigger role than entangling.

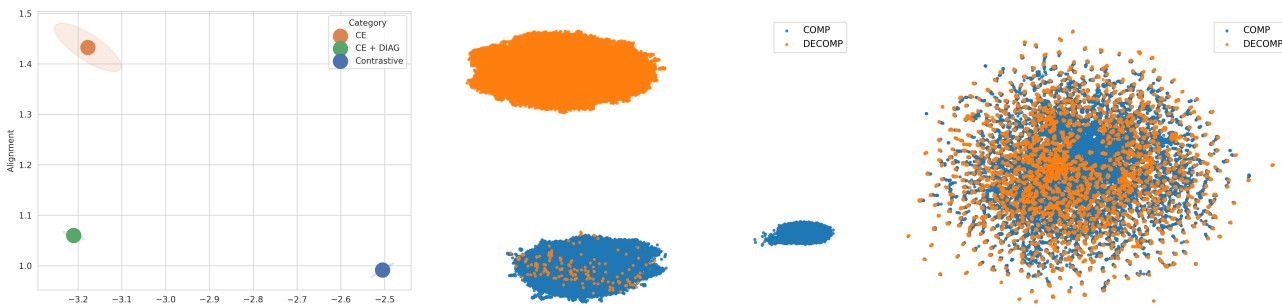

(a) Uniformity (↓) and Alignment loss (↓) analysis of BANYAN's representations using: CONTRASTIVE, CE and CE+DIAG.

(b) UMAP analysis of COMP and DECOMP embeddings for a 30k node entangled tree learned by CE w/o diagonal functions.

(c) UMAP analysis of COMP and DECOMP embeddings for a 30k node entangled tree learned by CE with diagonal functions.

Figure 5: Representation Analysis - UMAP Parameters in Appendix C.

**Diagonal Functions:** Perhaps the clearest benefits come from the introduction of the diagonalised composition and decomposition functions. These are bounded scalar values (sigmoid) multiplied with embeddings to mimic the time mixing blocks of SSMs (Gu & Dao, 2024). Hierarchically decaying in the influence of embeddings further down the structure through repeated application. This means that the representations are restricted to conform to the compression order it dictates, and we know from (Opper et al., 2023b) that the more we enforce this constraint the better our representations. Secondly, such simple message passing functions bias the representation space towards informative separability. There has to be some signal from which to perform reconstruction, and all the pressure is now on the representations. This is beneficial with the contrastive loss, but really shines when we combine them with cross entropy.

**Changing Objective:** Our more instructive finding is that cross entropy now outperforms the contrastive loss used by Opper et al. (2023b), contrary to the earlier result that the contrastive objective was critical. Figure 5 provides an analysis with Figure 5a showing uniformity vs. alignment metrics (↓) from Wang & Isola (2020). These measure (a) how evenly spread embeddings are (uniformity) and (b) the proximity of locally contextual composition embeddings to their globally contextualised counterparts (alignment). Success requires having low scores on both. We can see that the contrastive loss does well, while cross entropy achieves high uniformity, but not alignment without diagonal functions. This is further confirmed when we look at the UMAP (McInnes et al., 2018) analysis of BANYAN's embeddings with (5c) and without (5b) diagonal functions. Without diagonal functions, composition and decomposition embeddings largely occupy separate subspaces, failing to amortise over context. Introducing diagonal functions results in incredibly high overlap between the two, indicating effective amortisation. We believe that this is due to their simplicity and scaling constraints, restricting complex transformations during message passing. This *implicitly* forces representations to optimise the same useful qualities as the contrastive

loss, without its propensity for shortcut solutions (Robinson et al., 2021). As a result, we can switch objective without compromising representation quality.

## 8. Conclusion, Limitations and Future Work

We introduce BANYAN, a Self-Structuring AutoEncoder. BANYAN's focus on global, entangled structure and simplified message passing exploits the benefits of structured compositions inherent in language data. It is more effective and efficient than prior work from which we draw three central conclusions.

Firstly, explicitly modelling structured compositions is an effective inductive bias. Table 5 shows the parameters for the structured models versus the baselines. The structured models are far smaller, with tens or thousands of parameters instead of millions or billions. And nonetheless, BANYAN is still competitive across several metrics, indicating we have found an efficient learning procedure.

Secondly, we have not yet fully exploited the potential of the inductive bias. BANYAN still relies on greedy agglomerative clustering to induce structure. This is effective, but sub-optimal. Future work could learn the structure induction procedure. The type of structure models are exposed to impacts the quality of learnt semantic representations (Opper et al., 2023b). So if *how* we induce structure improves, the model should learn better representations.

Finally, good and cheap embedding models are useful for many applications. For example, the digital humanities need to organise corpora of ancient languages, making it easier for researchers to access texts they need. But these corpora are small, and these languages are unlikely to be present in pretraining corpora of larger models. BANYAN provides an efficient solution for producing representations for both these use cases and low resource languages and under represented communities more generally. To conclude, Banyan addresses the problem of efficient learning in low-resource settings.

## Acknowledgements

MO was funded by a PhD studentship through Huawei-Edinburgh Research Lab Project 10410153. We also wish to thank Henry Conklin, Seraphina Goldfarb-Tarrant, Vivek Iyer, Ivan Vegner, Sahil Verma, Su Kara, Ivan Titov and Edoardo Ponti for their valuable comments and feedback, throughout the research and development of this work. We'd also like to thank the ICML reviewers for their suggestions to help improve the paper. Finally, a special thank you goes to James Morrison, for helping to think through the ideas and providing invaluable feedback and edits to the manuscript.

## Impact Statement

This paper represents an effort to enable resource efficient embedding techniques. We hope that our model makes a positive step towards making AI research and technologies more fair, equitable and accessible.

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

## A. The $k$ and $u$ balance

The change to diagonal composition functions allows us to reduce the number of total parameters while maintaining performance. This is because the number of parameters is directly proportional to channel size $u$. We show ablations for this finding in Table 7. Our findings are similar to those of (Opper & Siddharth, 2024) the smaller the channel size the better the model performs, although in our case we keep things stable between seeds whereas before simplification caused issues with extreme instability during training. This is thanks to the new message passing functions.

Table 7: Performance Depending on $k$ and $u$ values using new functions. Scores are the average of four random seeds.

| $k$ | $u$ | Lex Score | STS Score |
|-----|-----|-----------|-----------|
| 4 | 64 | $47.83 \pm 0.2$ | $55.50 \pm 0.22$ |
| 8 | 32 | $47.41 \pm 1.0$ | $62.58 \pm 0.11$ |
| 16 | 16 | $48.01 \pm 1.1$ | $62.73 \pm 0.1$ |
| 32 | 8 | $47.65 \pm 1.1$ | $62.79 \pm 0.07$ |
| 64 | 4 | $48.48 \pm 0.7$ | $62.63 \pm 0.16$ |
| 128 | 2 | $48.53 \pm 1.33$ | $62.97 \pm 0.23$ |
| 256 | 1 | $49.15 \pm 0.6$ | $62.61 \pm 0.23$ |

## B. Hyperparameters:

We trained SELF-STRAE and BANYAN for 15 epochs (circa 15k steps and sufficient for convergence) using the Adam optimiser (Kingma & Ba, 2015), with a learning rate of 1e-3 for BANYAN and 1e-4 for SELF-STRAE using a batch size of 512. To process the graphs we used DGL (Wang et al., 2020). The GLOVE baseline was trained for 15 epochs with a learning rate of 1e-3, and a window size of 10. We used the official C++ implementation. ROBERTA medium was trained for 200,000 steps, (10% of which were used for warmup). We used a learning rate of 5e-5, and a linear schedule. Positional embeddings are relative key-query. We used the Transformers library to implement and train the model (Wolf et al., 2020). For SimCSE training, we used the default parameters and the official implementation for unsupervised ROBERTA training from Gao et al. (2021). For Sent2Vec we used their official implementation and recommend hyperparameters.

## C. UMAP Parameters:

For the UMAP visualisations, we set number of neighbours to 100, minimum distance to 0.3, the metric to cosine and local connectivity to 3. However, the same patterns can we observed through a wide array of hyperparameters, and when changing the metric to euclidean distance. The behavioural changes induced by the diagonal functions remain clear. We selected the above purely based on aesthetic preference for the resulting plots.

