# OpenReview forum: "Banyan: Improved Representation Learning with Explicit Structure"
_ICML.cc/2025/Conference — ICML 2025 poster_

### Official Review · Reviewer_Kous · 2025-02-19

**Overall Recommendation:** 4

**Summary:**

This paper introduces Banyan: a new recursive graph neural network for learning text representations in low-resource languages. This model extends previous work by building nested trees over sequences that share the same tokens. In Banyan the same tokens will have the same tree node, even if they come from different sequences. For scalability reasons, the trees are constructed from a batch of samples rather than from an entire dataset. Embeddings are learned from a simplified message-passing algorithm that traverses the trees both in bottom-up and top-down directions. Having nested trees provides multiple advantages, notably the reduction of duplicated nodes and multiple context representations within the same node. These advantages translate to strong semantic representations in both English (when compared to RoBERTa & GLOVE) and lower-resourced languages (when compared to XLM-R (fine-tuned), Llama 3.1 8B, Mistral Nemo 12B, MiniL12, and Paraphrase XLM-R).

## update after rebuttal

**Claims And Evidence:**

yes, the claims made in the submission are supported by clear and convincing evidence.

**Essential References Not Discussed:**

No essential reference missing to the best of my knowledge

**Experimental Designs Or Analyses:**

Experimental designs are sound. I checked their sentence and word level evaluation as well as their ablation study, and retrieval and classification tasks on English. I also checked their multi-lingual results.

**Methods And Evaluation Criteria:**

Yes, the proposed methods and evaluation benchmarks make sense for the application at hand.

**Other Comments Or Suggestions:**

In table 3, the last two columns should also bold the best numbers (Banyan @ 77.2 accuracy for sst-2 and Glove & RoBERTa @ 81 for MRPC)

**Other Strengths And Weaknesses:**

### Strengths

This paper introduces a novel recursive model that learns textual representations and its learning mechanism. The proposed architecture is novel and seems promising as it yields good results when compared to other more classical methods. In addition, the proposed method is very efficient: it requires very little training and has only 14 non-embedding parameters.

### Weaknesses

The task used to evaluate models is not immediately clear from the abstract or the introduction. While Semantic Textual Similarity - STS is mentioned in the introduction, it is not well explained what the objective is. The paper could gain clarity by explaining briefly what the goal is: having cosine similarities that map sentence pairs in a similar way to human judgements.

Table 3 shows results for retrieval and classification tasks from the beir and glue benchmarks, but very little information is given regarding the experimental setup and the result analysis. The paper would be stronger with detailed experimental design choices and result analysis on these benchmarks.

**Questions For Authors:**

No additional questions

**Relation To Broader Scientific Literature:**

The contribution of this paper is related to the STRAE model (Opper et al., 2023) published at EMNLP in 2023. This previous work is a similar, tree-based, sentence-level embedding method, that models compositional semantics with minimal data and model size requirements. Banyan differentiates itself by introducing entangled trees, which provides better performance and resource efficiency.

**Theoretical Claims:**

I did not check the correctness of any proofs or theoretical claims as I did not find any.

---

> ### Author Rebuttal · Authors · 2025-04-01
>
> Thank you for your review and feedback regarding the paper!
>
> The overall logic for our experiments is as follows:
>
> We care about whether we can create a resource efficient model by applying an inductive bias. This can be very useful for low resource languages. However, such languages do not have many high quality labelled datasets. The SemEval STS benchmark is the only wide ranging benchmark that we know off. What we want to do is take a high resource language (i.e. English) with a lot of labelled datasets and use that to establish whether we can learn high quality embeddings. STS provides one view of that. However, we should also make certain that Banyan succeeds at other tasks you might care about. The retrieval and classification tasks are there to demonstrate that the embeddings learned by Banyan capture multifaceted aspects of semantics. More concretely:
>
> Quora: This is about matching questions to answers. Crucial for applications like RAG and lets us see whether our embeddings capture the response relation.
>
> Arguana: This requires matching arguments to counter arguments. It lets us see whether our semantic space captures the notion of dialectical opposition.
>
> SST: Sentiment classification - does our representation space capture semantic polarity.
>
> MRPC: Paraphrase detection: does our representation space capture semantic equivalence.
>
> Moreover, these tasks utilise measures other than correlation, which provides a broader basis for measuring the success of embeddings. By demonstrating broad spectrum capabilities in English, and then showing similar trends but under a more limited evaluation in the low resource languages, we aim to show that Banyan a) learns effective representations b) can therefore provide a solution in case where other methods that require scale are inadequate.
>
>
> You are right that this could be better clarified and we will make sure to use the extra page if accepted to do so! Let us know if you have any further questions, and we look forward to engaging with you during the discussion period.

---

### Official Review · Reviewer_GCxG · 2025-03-11

**Overall Recommendation:** 4

**Summary:**

This paper studies the problem of learning semantic representations for language in low-resource settings. While word embeddings can be learned with little data, they are non-contextual; on the other hand, transformers can produce contextual embeddings but are data-hungry. In this work, the authors build on an existing architecture, that of Self-StrAE, and propose Banyan. The two main changes to Self-StrAE are that (1) in the downward pass, embeddings for spans with the same sequence of tokens are averaged following an entangled graph structure, and (2) neural models for embedding combination and decombination are replaced with simple diagonalized functions. The resulting model is very efficient in that training is very quick and inference can be done on CPU. The authors evaluate the model on various word-level and sentence-level tasks, first on English, and then on various low-resource languages, comparing it to word embeddings, Self-StrAE, and various transformer baselines trained on much more data. Banyan outperforms all baselines in most settings. Finally, they perform ablations and show that each proposed architectural change is beneficial.

**Claims And Evidence:**

- The main claim is that Banyan outperforms various baselines (word embeddings, Self-StrAE, and transformers) in representation learning for low-resource settings. This claim is well-supported by evidence, and the authors perform thorough evaluation on a very wide range of tasks across many languages.

- The other main claim in the paper is that the proposed modifications to Self-StrAE are beneficial (entangled graphs, diagonalized functions, replacing contrastive loss with cross entropy). This claim is also well-supported by thorough ablations.

- The main limitation of the experiments is that the non-neural baselines are relatively weak. In particular, there are many well-known sentence embedding approaches that are not cited or compared to (please see "Essential references not discussed"). The lack of such baselines undermines the claim that Banyan represents a breakthrough for producing representations in low-resource settings.

**Essential References Not Discussed:**

The main set of missing references (and baselines) is the large body of past work on sentence embeddings. While the method proposed in this paper is novel with respect to these past works in that it induces structure, it still competes with these sentence embedding methods as a way to produce sentence representations in low-resource settings.

The following are a few methods that are most related to their setting:
- [Arora, Liang, Ma (2017)] (https://oar.princeton.edu/bitstream/88435/pr1rk2k/1/BaselineSentenceEmbedding.pdf), and its extension [Ethayarajh (2018)] (https://aclanthology.org/W18-3012.pdf): at a high level, these methods take a weighted average of the word embeddings and apply an SVD
- [Pagliardini, Gupta, Jaggi (2017)] (https://arxiv.org/pdf/1703.02507): this method (sent2vec) extends the word2vec objective to sentences
- [Ruckle et al. (2018)] (https://arxiv.org/pdf/1803.01400): they show that averaging embeddings with a power mean (e.g. max pooling) and doing concatenation performs better than just taking the average.

In particular, the STS numbers reported in these papers are much higher than those of the GloVe baseline presented in this work.

**Experimental Designs Or Analyses:**

The experimental design (English experiments, multilingual evaluation, ablations) all seem sound. The main limitation of the results is the weak non-neural baseline (in particular, the lack of sentence embedding methods), as discussed below. The paper also lacks any non-neural baselines in the multilingual evaluations (Table 4), omitting the word embedding baseline present in the English experiments.

**Methods And Evaluation Criteria:**

- The method makes sense and the modifications to Self-StrAE are well-motivated. The efficiency of the method also makes it well-suited for low-resource settings.

- The benchmarks chosen are also reasonable in that they study (1) how well the representations match human judgments of semantic similarity, (2) how well the representations perform in retrieval, and (3) how well the representations perform for classification. Banyan performs well for (1) and (2) and is slightly worse than baselines for (3).

**Other Comments Or Suggestions:**

- Overall, the paper is very well-written, but section 5.4 is slightly harder to follow than the rest of the paper (but it is still clear enough).
- In my first pass through the paper, I didn't realize that Banyan used CE loss instead of contrastive loss. Maybe that is worth mentioning at the end of Section 4.
- In Section 4, I think it's also worth briefly describing how K and U relate to word embedding dimension, and what it means to have independent channels.

Typos:
- 163: ask -> asks
- 383: lightweigh -> lightweight
- 426: elegant. -> elegant,

**Other Strengths And Weaknesses:**

N/A

**Questions For Authors:**

- I am a bit confused that switching from C to CE loss on standard trees is better (Table 5), given that the Strae paper found the opposite conclusion. My guess is that C > CE for supervised StrAE, but CE > C for Self-StrAE, which makes intuitive sense because supervising intermediate nodes directly can be brittle if the trees are self-induced and slightly wrong. Is this the correct intuition?

- I am curious about the induced trees, whether they seem qualitatively reasonable, and whether they match human-designed trees (e.g. on the Penn Treebank). In particular, if the method does in fact recover reasonable trees, that would set it apart from previous sentence embedding methods.

**Relation To Broader Scientific Literature:**

The paper proposes a new architecture for structured representation learning which is novel, and it does a good job of positioning and motivating the proposed changes with respect to past work (Self-StrAE). However, it is somewhat unclear how this method relates to simpler existing approaches for sentence embeddings (please see the next section).

**Theoretical Claims:**

N/A

---

> ### Author Rebuttal · Authors · 2025-04-01
>
> Thank you for your extensive review and extremely helpful feedback!
>
> Other sentence embedding baselines:
>
> We appreciate you providing the missing references and will be sure to include them in the paper.
>
> Our key focus is to assess whether we can create an efficient method for learning representations for use with low resource languages. [1] Baseline Sentence Embeddings and [2] Unsupervised Random Walk both initialise their embeddings from pre-trained GloVe vectors trained on 840 billion tokens of text. This does naturally lead to better performance, but is only possible for super high resource languages like English, and not the low resource settings that we are primarily concerned with.  Power mean combination [4] also use pre-trained embeddings. Further the difference in performance [4] reports on sst is negligible. When we attempted to replicate their method we also found no meaningful change.
>
> Sent2Vec [3] reports several pretraining settings, the smallest of which is 900 million tokens with 700D vectors, which again is beyond low resource scale. However, the method is strong and certainly seems worth comparing to. We used the official implementation and followed their hyperparameter recommendations to train our own version.  We used our English Wikipedia subsample and at 256D for parity. Results are as follows:
>
> | Model    | STS-12 | STS-13 | STS-14 | STS-15 | STS-16 | STS-B | SICK  | SemRel | Score |
> |----------|--------|--------|--------|--------|--------|-------|-------|--------|-------|
> | Banyan   | 51.2 +- 0.007   | 69.1 +- 0.002  | 63.3 +- 0.004  | 73.2 +- 0.002  | 66.6 +- 0.002  | 61.5 +- 0.002  | 55.5 +- 0.003  | 61.6   +- 0.002 | 62.7 +- 0.001  |
> | Sent2Vec | 38.14 +- 0.29  | 51.37 +- 0.48  | 48.64  +- 0.09 | 67.28  +- 0.023 | 56.26 +- 0.06 | 53.39 +- 0.11 | 59.67 +- 0.02 | 51.47 +- 0.03 | 53.28 +- 0.11 |
>
> | Model    | Q N@1 | Q N@10 | Q R@1 | Q R@10 | A N@1 | A N@10 | A R@1 | A R@10 | SST   | MRPC |
> |----------|-------|--------|-------|--------|-------|--------|-------|--------|-------|------|
> | Banyan   | 57.83 +- 0.04 | 65.78 +- 0.05 | 50.19 +- 0.08 | 75.80 +- 0.18 | 13.21 +- 0.25 | 29.28 +- 0.11  | 27.41 +- 0.68 | 49.60 +- 0.52  | 79.51 +- 0.16 | 77.2 +- 0.27 |
> | Sent2Vec | 36.12 +- 0.21 | 43.26 +- 0.15  | 31.33 +- 0.21 | 52.38 +- 0.05  | 9.6 +- 0.31  | 23.24 +- 0.15  | 9.6 +- 0.31  | 39.73 +- 0.89  | 76.53 +- 0.98 | 81 +- 0.0  |
>
> | Model    | Simlex | WordSim S | WordSim R | Score  |
> |----------|--------|-----------|-----------|--------|
> | Banyan   | 16.57 +- 0.02  | 63.25 +- 0.03     | 69   +- 0.01    | 49.61 +- 0.02 |
> | Sent2Vec | 28.88 +- 0.42 | 68.32 +- 1.26    | 54.49  +- 1.51   | 50.56 +- 0.79 |
>
> Sent2Vec is stronger than our original WE baselines, though Banyan generally retains an edge.
>
> Multilingual Word Embedding Baseline:
> We only compare to large pretrained models here because we hope we already prove Banyan’s utility compared to other methods you could easily train from scratch in our initial experiments. Nevertheless we have trained sent2vec for a few of the languages and find similar trends to the results on English:
>
> Afrikaans:
> Banyan: 78.68 +- 0.30
> Sent2Vec: 73.36 +- 0.55
>
> Telugu:
> Banyan: 71.13 +- 0.91
> Sent2Vec: 68.58 +- 0.58
>
> Spanish:
> Banyan: 60.95 +- 0.76
> Sent2Vec: 55.15 +- 0.54
>
> Sent2Vec requires a sentence tokenised corpus as input and finding and then running tokenisers for all the languages is proving to be quite time consuming and in some cases challenging. We are however happy to work on filling out a sent2vec baseline for all languages if you strongly feel this would improve the paper.
>
> Suggestions: Thank you for pointing these out, all are good improvements and we will edit accordingly!
>
> C > CE: It depends what you mean by brittle?  Using the contrastive loss does lead to consistent performance. However, it can lead to certain unfavourable behaviour whereby tokens like ‘the’ are excessively pushed away from all other embeddings which introduces oddities in the structure. However, C is still superior to CE because it adds a whitening effect, which is vital for tasks like STS. To retain whitening while using CE we had to switch to the diagonal functions. These are simple and therefore need easily separable embeddings to still do well on reconstruction.
>
> The Trees:  To an extent they do seem reasonable, although our merge algorithm is context free which is a limitation on the types of structure. It is maybe best to think of them as akin to running BPE all the way till the whole sequence is compressed. Qualitatively we do find that this leads to some reasonable patterns, and consistent behaviours such as segmenting phrases can be observed. We would be happy to include some examples in the appendix in the final version!
>
> Thank you for your constructive feedback and time taken with the paper, we look forward to engaging with you during the rebuttal period!  If you feel like our rebuttal has addressed your concerns please consider raising your score.

---

> > ### Comment · Reviewer_GCxG · 2025-04-04
> >
> > Thanks for the detailed reply! I find the additional experiment convincing and have raised my score.
> >
> > If you have time, I would be curious about the [Ruckle et al. (2018)](https://arxiv.org/pdf/1803.01400) baseline as well because it is quite simple. Instead of taking a simple average over the word embeddings, it takes the power-mean for various values of p (where p=infty corresponds to max pooling), and then concatenates these embeddings together. This simple method also has strong performance for semantic similarity tasks and should be doable in low-resource settings since it only requires word embeddings.

---

> > > ### Author Response · Authors · 2025-04-09
> > >
> > > We ran some evaluations with the best configuration from Ruckle et al. [-inf, 1, inf] and found some mixed results:
> > >
> > > | Model         | STS-12         | STS-13         | STS-14         | STS-15         | STS-16         | STS-B          | SICK           | SemRel        | Score          |
> > > |---------------|----------------|----------------|----------------|----------------|----------------|----------------|----------------|---------------|----------------|
> > > | GloVe         | 39.00 +- 0.57  | 41.61 +- 0.19  | 39.31 +- 0.18  | 51.06 +- 0.35  | 45.14 +- 0.14  | 48.40 +- 0.07  | 52.80 +- 0.04  | 42/37 +- 0.13 | 44.96 +- 0.1   |
> > > | + Power Mean  | 35.35 +- 0.79  | 42.44 +- 0.83  | 39.72 +- 0.42  | 47.93 +- 1.09  | 43.08 +- 0.76  | 50.73 +- 0.03  | 51.17 +- 0.06  | 38.81 +- 0.25 | 43.65 +- 0.65  |
> > >
> > >
> > > It seems helpful in some cases but quite detrimental in others, overall leading to a slightly worse score...
> > >
> > > We will keep experimenting to see if it can be improved, but overall we don't expect the picture to change much based on the results reported by Ruckle et al in Table 2. The performance increase seems very slight - and insufficient to bridge the gap even to sent2vec's performance level.
> > >
> > > Thanks again for your helpful feedback and constructive review, it provided some really useful context to inform our work!

---

### Official Review · Reviewer_PYAv · 2025-03-16

**Overall Recommendation:** 4

**Summary:**

The paper proposes BANYAN, a graph-based autoencoder that learns sentence representations by explicitly encoding hierarchical structures. It extends a prior structured model (SELF-STRAE) in two major ways: 1) Entangled Trees: Instead of building a separate tree per sentence, the model merges identical token spans across all sentences (in a batch) into shared nodes. Each node can thus gather training signals from multiple contexts, rather than duplicating the same span across different sentences. This “entangling” reduces memory usage (fewer total nodes) and helps prevent conflicting training signals. 2) Diagonalized Message Passing: The composition and decomposition functions that build and break down embeddings along the tree are replaced with tiny diagonal gating operations. Rather than a full matrix multiply, each dimension is scaled by a learned scalar in [0, 1], allowing the model to control how much information from each child flows upward or downward. This sharply cuts parameter count (just 14 scalars beyond the embeddings) and enforces a rigid “compression order” over the tree. Training uses a cross-entropy reconstruction of the original sentence, ensuring that the root embedding and each intermediate node carry enough information to decode their children. The structure itself is induced via a greedy merging algorithm that repeatedly combines the most similar pairs of embeddings, forming a tree bottom-up. By reusing nodes across sentences, BANYAN effectively averages how each repeated phrase is used in different contexts.

BANYAN is tested mainly on semantic textual similarity (STS) tasks at word and sentence level, plus a few classification and retrieval tasks. Key findings are that, despite having very few non-embedding parameters, Banyan matches or beats larger transformer models on many STS benchmarks. It excels especially in low-resource languages, where big multilingual transformers often struggle without abundant data.

**Claims And Evidence:**

The paper’s core claim is that explicit structure and minimal gating can yield strong, data-efficient embeddings. Results on various English and multilingual STS sets support this, showing BANYAN is both much smaller than conventional models and still highly competitive. Another claim is that merging identical spans across sentences removes duplication and leverages repeated phrases more effectively. This is demonstrated by the model’s consistent gains over the previous approach that used one tree per sentence, as demonstrated in ablations.

**Essential References Not Discussed:**

None I can think of.

**Experimental Designs Or Analyses:**

The experiments compare BANYAN to relevant baselines (both structured and transformer-based) on unsupervised semantic similarity tasks, plus a few downstream evaluations. The training setup (e.g., matching embedding sizes across models, consistent optimizer settings) is well-justified. The ablation study (entangled vs. standard trees, diagonal vs. full matrices, contrastive vs. cross-entropy) clarifies each modeling choice’s impact. No obvious flaws in methodology were detected.

**Methods And Evaluation Criteria:**

Banyan is an unsupervised method. The primary measure of success is whether the learned embeddings rank sentence pairs in alignment with human-rated similarity.
It also checks retrieval metrics (e.g., NDCG, Recall) and downstream classification performance with a frozen encoder to assess practical utility. Overall, the evaluation makes sense.

**Other Comments Or Suggestions:**

line 163: ask -> asks

**Other Strengths And Weaknesses:**

Strengths:

* Original: Entangled composition across multiple sentences is novel and addresses false negatives or duplicate merges.
* Efficiency: The model relies on only ~14 non-embedding parameters yet can rival large transformers, especially in low-resource contexts.
* Clarity: The paper is logically organized and methodologically transparent (ablation results, multilingual experiments). It is really well written.

Weaknesses:

There are no clear weaknesses. The paper supports all its claims quite well. One could point out that the model goes against the obvious trend of training larger models on more data, and will likely not play a big role in the future. But I wouldn't consider that a real weakness.

**Questions For Authors:**

I don't have questions that would change my evaluation.

**Relation To Broader Scientific Literature:**

Banyan follows a line of structured representation learning models but innovates by entangling repeated spans across sentences and using diagonal gating in the composition functions. This ties into prior work on compositionality, unsupervised parsing, and efficient RNN gating, yet it is unique in merging identical spans globally for more efficient, context-rich representations. The discussion references all key works I can think of.

**Theoretical Claims:**

The paper doesn’t present formal proofs but makes some conceptual claims, for example, that diagonal gating enforces a strict “compression order” over the tree and that averaging entangled nodes (batch-wise) is an unbiased estimator of their global context. These claims appear logically consistent with standard ideas in gating (e.g. decaying influence across multiple composition steps) and stochastic training. No glaring theoretical issues were found.

---

> ### Author Rebuttal · Authors · 2025-04-01
>
> Thank you for your feedback, and positive appraisal of our work! We will make sure to fix the typo - thanks for pointing that out!
>
> Regarding the trend of bigger models and more data, this is definitely the way things are moving, but it also leaves a lot of languages behind and limits who can participate in research. Efficient solutions can help bridge that gap.

---

> > ### Comment · Reviewer_PYAv · 2025-04-07
> >
> > I completely agree. This is good work and should find its way into the conference.

---

### Decision · Program_Chairs · 2025-05-01

**Decision:**

Accept (poster)

**Comment:**

This paper introduces Banyan, a graph-based autoencoder for learning sentence representations, particularly targeting low-resource languages. Banyan builds on prior work (Self-StrAE) by incorporating "entangled trees" (sharing identical spans across sentences in a batch) and using minimal gating parameters, resulting in a model with few non-embedding parameters. The efficiency and strong performance demonstrated on Semantic Textual Similarity (STS) and retrieval tasks, especially in low-resource settings compared to large transformer models, were recognized and appreciated by all reviewers.

The review process highlighted one major concern regarding the lack of comparison to relevant simple sentence embedding baselines (like Sent2Vec) suitable for low-resource scenarios and related work in general as it largely goes from word embeddings to pre-trained models. The authors provided a rebuttal, presenting new experimental results comparing Banyan against Sent2Vec (trained under comparable conditions). These results showed Banyan maintaining an edge. One other line of work worth mentioning (See "Paraphrastic Representations at Scale" for instance and original paper from 2015) first showed the effectiveness of averaging embeddings for textual similarity in transfer learning which has been shown to be competitive and outperform large pre-trained neural networks even today - just like Banyan. Especially with the rise of LLMs, assuming only a (non-parallel) corpus to learn embeddings can be weakened since it is easy enough to get some form of weak supervision with parallel data. This is because there is lots of translation data and LLMs can also translate themselves without being explicitly trained.

For the final version, the authors must incorporate the new baseline results (Sent2Vec, etc.) and analysis discussed in the rebuttal. They must also follow through on their commitment to enhance clarity. Addressing these points will strengthen the final paper.